# Is DFR for Soft Biometrics Prediction in Unconstrained Images Fair and Effective?

**Udaybhan Rathore and Akshay Agarwal**
Department of Data Science and Engineering
IISER Bhopal, India
{udaybhan19, akagarwal}@iiserb.ac.in

## Abstract

Face being a nonintrusive recognition modality, is an ideal candidate for identifying criminals. The modality is not only related to identity but can also extract several other important features such as age, race, and gender. In this preliminary research, we have collected a novel unconstrained face recognition dataset using mobile phones. On the collected dataset, we have evaluated the current state-of-the-art (SOTA) deep face recognition (DFR) algorithms for face attribute identification. This research aims to identify whether current algorithms are effective for the task or biased towards their training set. The results suggest that the current technology is not effective in identifying face attributes when the images are captured in unconstrained environments. For example, deep face networks yield the best F-1 score of $0.43$ when asked to predict gender on the collected dataset.

## 1 Indroduction

face recognition algorithms have shown tremendous success even surpassing human-level performance on some semi-constrained datasets Sun et al. (2015); O'Toole et al. (2007). Face modality is equipped with several other soft modalities, such as race, gender, and age. These soft modalities, if correctly identified, can limit the search space to identify a possible suspect of a crime. For example, suppose we have to identify a criminal in a dataset containing millions of identities of different races and gender. In that case, the possible hash function mapping the true race and gender can significantly boost the identification performance by only searching in the relevant race and gender individuals. However, we assert that the success of deep face recognition (DFR) algorithms on laboratory-captured datasets might not be transferred to the real world. For example, Dammak et al. (2021); Hsu et al. (2021) have showcased that the age and gender detection classification drops significantly when the images are captured in unconstrained environments. The studies are conducted on European, caucasian, and American ethnicities, and no study or few studies has worked on Indian ethnicity face images. Therefore, in this research, we have conducted a preliminary study using the Indian ethnicity face dataset captured in the real world and unconstrained settings. In brief, the contributions of this research are (i) real-world face recognition and face attribute analysis dataset and (ii) detailed experimental analysis concerning deep face recognition algorithm equipped with several face detectors. The analysis reveals that the current deep face recognition algorithms are not effective in identifying age and gender attributes. Apart from being an unconstrained dataset, the prime reason for such ineffectiveness can be the ethnicity bias of these algorithms.

## 2 Proposed Dataset

While face attribute identification is critical, it is hard to install a surveillance camera at every possible location in the world. Thanks to the advancement of mobile hardware, every mobile device is now equipped with a camera that can capture images of a person from even a long distance (say 10 meters). Utilizing this potential, we have collected a video-based face dataset of Indian individuals in the unconstrained real world. The subjects between the mobile camera and the subject have been kept fixed at 10 meters. We collected videos of 30 subjects (15 female and 15 male) where no special instructions were given, and they were free to perform natural tasks. The age range of the subjects is between 19 and 34, with an average of 22.36 years. The videos are captured at a frame rate of 30.01 frames/rate, with a frame width and height of $1920 \times 1080$ pixels, and last for 5-7 seconds. For experiments, we randomly selected 15 frames from a video for face analysis experiments.

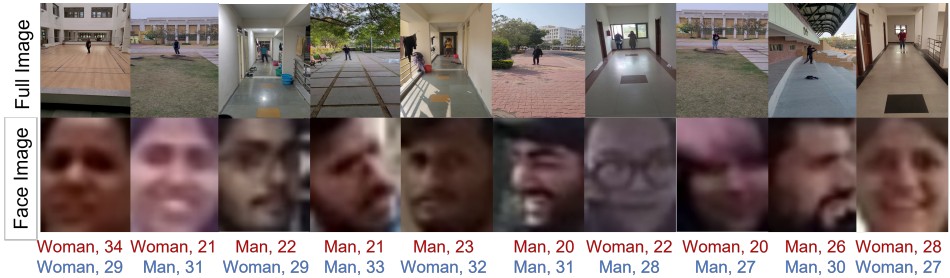

Figure 1: Full images and cropped face region samples from our collected dataset. Along with images, we have mentioned the true gender and age of each of the samples and the predicted age and gender by the deepface model. The SOTA models are highly ineffective on Indian ethnicity subjects in identifying age and gender. Red and blue represent the true and predicted values, respectively.

| Predicted ⟶ True ↓ | RetianceFace | | MTCNN | |
|---|---|---|---|---|
| | M | F | M | F |
| M | 210 | 0 | 210 | 0 |
| F | 215 | 25 | 228 | 12 |
| | SSD | | OpenCV | |
| M | 202 | 8 | 202 | 8 |
| F | 240 | 0 | 233 | 7 |

(a)                     (b)

Figure 2: (a) Confusion matrix for gender classification. (b) Age identification with different tolerance limits. The experiments are performed on a total of 450 images of 30 subjects using varying face detectors and reflect their ineffectiveness in classifying gender and age attributes.

# 3   EXPERIMENTS AND RESULTS

To perform the face attribute analysis, i.e., age and gender, we have used the deepface library Serengil & Ozpinar (2020). The experiments are performed using various face detector backends such as MTCNN, OpenCV, SSD, and RetinaFace. A few samples of our dataset collected from a 10 meters distance are shown in Figure 1 reflecting the challenging nature of the proposed dataset. It is reflected through the age prediction analysis that the current SOTA deep face models coupled with different backends are not effective for age estimation. The detailed analysis of age estimation is also in Figure 2. We have performed the age classification with different tolerance limits. The tolerance limits $k$ represent that if the predicted age is within $\pm k$, then we classified its correctly predicted age else incorrect age estimation. As we keep increasing the tolerance value, the accuracy of age prediction increases. Similarly, the Table shown in Figure 2 (a) showcases that while the majority of the face analysis algorithms are effective in classifying males, it is highly ineffective for females. It shows the biased behavior of the deep face recognition models.

We have now conducted experiments with a face attribute classification networks Roy (2020) trained on face datasets, namely UTKFace Song & Zhang (2017) and FairFace Karkkainen & Joo (2021), having balanced age, gender, and ethnicity images. The model trained on the UTKFace dataset yields the F-1 score of $0.39$, reflecting that even the models trained on a balanced dataset are ineffective under the unconstrained nature of the dataset. Similar to the model trained on UTKFace, the model trained on Fairface is able to correctly predict the age on $200$ images only out of $450$ within a tolerance of $10$ years which is significantly lower than the deepface recognition system.

# 4   CONCLUSION

In this preliminary research, we have collected the Indian ethnicity mobile dataset using 30 subjects at a 10-meter distance from the camera. The extensive experimental analysis concerning the identification of soft biometrics reveals that the current deep face recognition models are highly biased towards males and are ineffective for age estimation. In the future, we plan to extend our dataset and propose a novel, de-biased gender classification, and better age estimation network.

URM STATEMENT

The authors acknowledge that the first author of this work meets the URM criteria of the ICLR 2023 Tiny Papers Track.

DATASET ETHICAL STATEMENT

We want to highlight that the proposed dataset is collected by following strict ethical guidelines. Further, the subjects were informed that their biometric images are getting collected for research purposes and will not release any information related to identity. The dataset will be released to the research community after signing the license agreement and will preserve the privacy of the subjects in the best possible manner.

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
