# OpenReview forum: "Is DFR for Soft Biometrics Prediction in Unconstrained Images Fair and Effective?"
_ICLR.cc/2023/TinyPapers — Submitted to Tiny Papers @ ICLR 2023_

### Official Review · Reviewer_mBVo · 2023-03-28

**Confidence:** 5

**Summary Of Contributions:**

TL;DR: The paper investigates face recognition and attribute analysis of Indian individuals captured in uncontrolled settings using mobile devices. The authors find that current deep face recognition models are ineffective in age estimation and demonstrate gender bias towards males, with poor performance in the classification of females. They also highlight that the success of face recognition algorithms in semi-constrained datasets may not necessarily translate to real-world scenarios.

**Rating:**

Great Start (GS): a submission which meets some of the reviewing criteria but has room for improvement

**Strengths And Weaknesses:**

## Strengths:

1. The authors curated a real-world dataset for Indian ethnicity individuals in unconstrained settings at a distance.
2. The paper presents a detailed experimental analysis of deep face recognition algorithms and highlights the need for unbiased models for accurate face attribute analysis.

## Weakness:

1. Though the paper emphasizes face as a nonintrusive modality, it is considered intrusive. According to GDPR, Using biometric data such as facial images is very restricted under the GDPR (Article 9).
2. Though the paper claims, there is a lack of images from Indian ethnicity, it can be found in the UTKFace dataset and gender information in the Adience dataset.
3. Please consider evaluating works by [1] and [2] in your future study.
4. A sample size of 30 is a bit limiting to conducting the study. Please consider increasing the sample size and using other available datasets such as UTKFace and the Adience dataset.

## References:

1. Abdolrashidi, Amirali, Mehdi Minaei, Elham Azimi, and Shervin Minaee. "Age and gender prediction from face images using attentional convolutional network." arXiv preprint arXiv:2010.03791 (2020).
2. Levi, Gil, and Tal Hassner. "Age and gender classification using convolutional neural networks." In Proceedings of the IEEE conference on computer vision and pattern recognition workshops, pp. 34-42. 2015.


**Suggested Changes:**

1.  please correct a possible typo  suscept->suspect
2.  A sample size of 30 is a bit limiting to conducting the study. Please consider increasing the sample size and using other available datasets such as UTKFace and the Adience dataset.

---

> ### Author Response · Authors · 2023-04-17
> **Response**
>
> $\textbf{Ethical Concern:}$ Thank you for raising this concern. We appreciate your point of view regarding the use of facial images in biometric data, and we understand that the General Data Protection Regulation (GDPR) imposes strict regulations on the collection, processing, and storage of personal data. Therefore, we want to highlight that the proposed dataset is collected by following strict ethical guidelines. Further, the subjects were informed that their biometric images are getting collected for research purposes and will not release any information related to identity. The dataset will be securely released to the biometrics community after signing the license agreement and will preserve the privacy of the subjects in the best possible manner. We have now added the ethical statement in the updated paper.
>
> $\textbf{Other Dataset:}$ Thanks for pointing out the datasets. We will take a look at these datasets and will include them in our future studies. However, one important thing which we would like to highlight is that the proposed dataset does not only aim to handle Indian ethnicity but also aim to evaluate the effectiveness of existing deepface models for the images captured at unconstrained and large stand-off distance between subjects and camera. The above-mentioned datasets do not possess this feature (large unconstrained standoff distance).
>
> Additionally, our primary objective is to create a long-range distance dataset using a mobile phone, which is now accessible to almost everyone and is cost-effective to be deployed by financially limited institutions.
>
> $\textbf{Future Evaluations:}$ Thank you for bringing these two research papers to our attention. We will definitely incorporate their findings into our future research on the proposed dataset.
>
> However, based on the suggestion of the reviewer, we have now conducted an additional experiment with a face attribute classification network trained on face datasets, namely UTKFace [3], having balanced age, gender, and ethnicity images. The model trained on the UTKFace dataset [4] yields the F-1 score of 0.39, reflecting that even the models trained on a balanced dataset are ineffective under the unconstrained nature of the dataset. Interestingly, the performance is lower than the deepface model used in the paper, yielding an F-1 score of 0.43 for gender prediction. Further, when the UTKFace model is used for the age prediction with a tolerance limit of 10, the model correctly classifies 317 images, whereas the deepface successfully predicts 400 images.
>
> We have also evaluated the face attribute prediction model [6] trained on another ethnicity-balanced dataset, namely Fairface [5]. Similar to the model trained on UTKFace, the model trained on Fairface is able to correctly predict the age of 200 individuals only within a tolerance of 10 years which is significantly lower than the deep face recognition system used in the paper.  We also analyzed the race prediction on our proposed dataset; we found that the model predicted most of the faces to be of White, Black, and Asian races. However, it was only accurate in predicting the correct race 96 times out of 450, showcasing the biased behavior of existing models.
>
>
> $\textbf{Dataset Extension:}$ Thank you for the suggestion. Currently, we are in the process of expanding our dataset size from 30 to 100-150 subjects and at varying standoff distances, which we have also pointed out in the future work of the paper.
>
> We will be more than happy to provide further information if needed to resolve the concerns. We want to thank all the reviewers for their constructive feedback.
>
> [3] https://www.kaggle.com/datasets/jangedoo/utkface-new
>
> [4] https://github.com/abr-98/Face_data_based_deeplearning
>
> [5] https://github.com/dchen236/FairFace
>
> [6] https://github.com/dchen236/FairFace/blob/master/predict.py

---

### Official Review · Reviewer_5GzM · 2023-04-02

**Confidence:** 5

**Summary Of Contributions:**

This paper proposes a new facial dataset with unconstrained images taken in the real world. Following that, it also conducts experimental analyses to determine whether current facial recognition algorithms are fair and unbiased for the tasks of age and gender prediction.

**Rating:**

Needs Clarification (NC): a submission which does not meet the reviewing criteria and needs clarification for its described problem or solution

**Strengths And Weaknesses:**

**Strengths:**

- The authors test multiple different facial recognition backbones as part of their experiments.

**Weaknesses:**

- (**Note to PCs**) I have concerns about the ethical considerations of this work. The authors have proposed and collected data, including facial images, for this paper. Unfortunately, there is no clarification provided on whether consent was obtained from the subjects.
  - This is further exacerbated, given that the data is being used to train models to estimate the gender and age of the subjects.

- The authors issue a blanket statement denouncing most facial recognition models as biased (especially towards Indians). However, they fail to make any comparisons with models trained on UTKFace [1] or FairFace [2], datasets that are known to incorporate different races and identities. In fact, the authors fail to show any quantitative results that back their claims of models being biased.

- There also seem to be no experiments with current state of the art models in gender and age estimation.

````
[1] Krishnan, A., Almadan, A., & Rattani, A. (2020, December). Understanding fairness of gender classification algorithms across gender-race groups. In 2020 19th IEEE International Conference on Machine Learning and Applications (ICMLA) (pp. 1028-1035). IEEE.
[2] Kärkkäinen, K., & Joo, J. (2019). Fairface : Face attribute dataset for balanced race, gender, and age. arXiv preprint arXiv:1908.04913.
````

**Suggested Changes:**

- Leaving ethical concerns aside, this paper deals with complex social issues pertaining to the invasion of privacy and data usage. The wording of the paper (especially for identifying people in public places) leaves a lot to be desired. At the very least, I would suggest adding some mention and add references to how bad actors can misuse these technologies for personal gain.

- Based on reading the paper, the authors have mentioned the data being collected with a handheld mobile camera recording people in the real world in unconstrained settings. I sincerely hope appropriate permissions were sought before collecting data and making predictions.

- The experiments in the paper show that the backbones finetuned on the dataset provided by the authors are ineffective for age/gender estimation. However, there exist subtle differences between generic and fine-grained image analysis. The backbone networks (part of the DeepFace library) used by the authors pertain to the latter, while the dataset used by the authors is closer to the former. Appropriate measures (like using different models or backbones or changing the architecture entirely) should be used for a fairer comparison.
  - The authors have mentioned developing a better, de-biased model, which may be a step in this direction.

---

> ### Author Response · Authors · 2023-04-17
> **Response**
>
> Thanks for raising this important concern. We want to highlight that the proposed dataset is collected by following strict ethical guidelines. Further, the subjects were informed that their biometric images are getting collected for research purposes and will not release any information related to identity. The dataset will be securely released to the biometrics community after signing the license agreement and will preserve the privacy of the subjects in the best possible manner. We have now added the ethical statement in the updated paper.
>
> Based on the suggestion of the reviewer, we have now conducted an additional experiment with a face attribute classification network trained on face datasets, namely UTKFace [3], having balanced age, gender, and ethnicity images. The model trained on the UTKFace dataset [4] yields the F-1 score of 0.39, reflecting that even the models trained on a balanced dataset are ineffective under the unconstrained nature of the dataset. Interestingly, the performance is lower than the deepface model used in the paper, yielding an F-1 score of 0.43 for gender prediction. Further, when the UTKFace model is used for the age prediction with a tolerance limit of 10, the model correctly classifies 317 images, whereas the deepface successfully predicts 400 images.
>
> We have also evaluated the face attribute prediction model [6] trained on another ethnicity-balanced dataset, namely Fairface [5]. Similar to the model trained on UTKFace, the model trained on Fairface is able to correctly predict the age of 200 images out of 450 within a tolerance of 10 years which is significantly lower than the deep face recognition system used in the paper.  We also analyzed the race prediction on our proposed dataset; we found that the model predicted most of the faces to be of White, Black, and Asian races. However, it was only accurate in predicting the correct race 96 times out of 450, showcasing the biased behavior of existing models.
>
> [3] https://www.kaggle.com/datasets/jangedoo/utkface-new
>
> [4] https://github.com/abr-98/Face_data_based_deeplearning
>
> [5] https://github.com/dchen236/FairFace
>
> [6] https://github.com/dchen236/FairFace/blob/master/predict.py

---

### Official Review · Reviewer_6W7h · 2023-04-02

**Confidence:** 4

**Summary Of Contributions:**

The paper presents a new dataset of individuals identifying as of Indian ethnicity for face attribute identification. It also presents the results of the evaluation of a soft biometrics predictor using the proposed dataset.

**Rating:**

Great Start (GS): a submission which meets some of the reviewing criteria but has room for improvement

**Strengths And Weaknesses:**

The paper presents an analysis on the performance of an existing soft biometrics recognition network when tested in the wild on people identifying as of Indian ethnicity. The analysis reviews the known and important issue of bias towards the data distribution seen during training, that can heavily affect performance when tested on different data.
To do so, a new video dataset is collected using mobile phones and fixed acquisition distance in unconstrained environments.
While this paper presents only an analysis of existing methods, the authors mention that future work will include a novel, de-biased gender classification and age estimation networks.

The paper also presents some weaknesses.
- Some important details of the proposed dataset are missing, such as the number of M/F subjects and their age distribution, the video and image resolution, and whether the dataset is expected to be published.
- Four different face detection methods are tested, while only one soft biometrics predictor is evaluated. Given that the focus of the paper is on soft biometrics recognition performance, I would have expected the evaluation of different recognition methods rather than different face detection models.
- According to Serengil & Ozpinar (2020), their model expects the face to be detected by OpenCV and horizontally aligned, but I didn't find this alignment step mentioned in the text.
- The Table in Figure 2(a) is wrongly formatted (MTCNN and OpenCV span one column instead of 2) and it could contain both relative and absolute values.
- The English level is good in general, but the text contains some minor typos (e.g. suscept instead of suspect)
- Overall, I'm not totally convinced by the motivations provided in the abstract and the introduction. They seem to be a specific use case (surveillance/policing), which also raises ethical questions, while soft biometrics can be used in a variety of different tasks.

I also wonder if the degraded performance of LightFace is caused by the ethnicity of the participants or by the very low resolution of their face. As future work, the authors could consider to extend their dataset with recordings at different distances to assess how much the face resolution affects the overall performance.

**Suggested Changes:**

I suggest to include additional information about the proposed dataset in the text (an Appendix section can be added if needed), proofread the text for typos, and improve the formatting of the Table in Figure 2(a).

I also encourage the authors to add an "Ethics Statement" section, as recommended in the [Author Guide](https://iclr.cc/Conferences/2023/AuthorGuide), to discuss the adherence of their paper to the [ICLR Code of Ethics](https://iclr.cc/public/CodeOfEthics).
In particular, I think that two points should be discussed: the collection of the dataset and the publication approval of the participants, and the potential consequences that effective in-the-wild soft biometrics predictors could have on the society if used for surveillance/policing.
The ethic statement does not count toward the page limit, but should not be more than 1 page.

---

> ### Author Response · Authors · 2023-04-17
> **Response**
>
> We would like to thank the reviewers and meta-reviewers for appreciating the efforts taken to capture the dataset, along with highlighting the important issue of bias in soft-biometrics identification. While in the literature, several studies have been conducted for face recognition, and limited work has been done towards understanding the biasness in evaluating soft biometrics, including in the Indian context.
>
> $\textbf{Dataset Details:}$ We would like to thank the reviewer for highlighting this point. Here is the information R1 requested to provide: To avoid any biasedness in the evaluation, we have experimented with an equal gender ratio, i.e. 15 subjects are male, and 15 subjects are female. The age range of the subjects is between 19 and 34, with an average of 22.36 years. The videos are captured at a frame rate of 30.01 frames/rate, with a frame width and height of 1920×1080 pixels, and last for 5-7 sec. The videos are captured in unconstrained indoor and outdoor environments where no special instructions are provided to the subjects and users are free to perform any natural motions. Further, the proposed dataset will be released to the public for research purposes, and one can receive the dataset after signing the license agreement.
>
> $\textbf{Face Detection and Bias:}$ The motivation for utilizing the different face detection models is that the face is the critical component, and the effectiveness of the detection model significantly impacts the recognition performance[1-2]. Therefore, in this preliminary study, we evaluated several face detection models.
>
> Although, based on the suggestion of the reviewer, we have now conducted an additional experiment with a face attribute classification network trained on face datasets, namely UTKFace [3], having balanced age, gender, and ethnicity images. The model trained on the UTKFace dataset [4] yields the F-1 score of 0.39, reflecting that even the models trained on a balanced dataset are ineffective under the unconstrained nature of the dataset. Interestingly, the performance is lower than the deepface model used in the paper, yielding an F-1 score of 0.43 for gender prediction.
>
> Further, we have also evaluated the face attribute prediction model [6] trained on another ethnicity-balanced dataset, namely Fairface [5]. Similar to the model trained on UTKFace, the model trained on Fairface is able to correctly predict the age on 200 out of 450 images within a tolerance of 10 years which is significantly lower than the deep face recognition system used in the paper, showcasing the biased behavior of existing models.
>
> $\textbf{Alignment:}$ Thank you for the clarification. In their paper, Serengil & Ozpinar (2020) mention that OpenCV is the default detector used in their model, but they also mention other detectors that can be used. We have passed both aligned and raw frames as input for the analysis; however, no significant variation has been observed in the performance of face attribute analysis.
>
> $\textbf{Formatting Error:}$ We appreciate you bringing this mistake to our attention. We have updated this in the updated version of the submitted paper.
>
> $\textbf{Soft Biometrics:}$ We agree with R1's perspective that the soft biometrics model has diverse potential uses, such as aiding public event organizers, healthcare, and retailers in understanding their customer base and tailoring their marketing efforts accordingly. However, our primary focus is on developing a system for surveillance and security purposes, as we have observed that suspects often attempt to evade surveillance cameras by concealing their faces. As a result, we are specifically motivated to develop a dataset that is unconstrained and features subjects standing at a distance from the camera without looking directly at it. If we are able to create a biometric system that can achieve high accuracy on this type of dataset, it would be tremendously beneficial to society.
>
> $\textbf{Bias Factor:}$ Thank you for your insightful review. We agree that the degraded performance of LightFace could be attributed to a variety of factors, including both the ethnicity of the participants and the low resolution of their faces. To test this intuition, we have evaluated the model trained on fair face for ethnicity evaluation to see whether ethnicity has any impact or its only image resolution on soft biometrics prediction. The Model predicted most of the faces are of White, Black, and Asian races. Only 96 times out of 450 predict the correct race, which implies that the current dataset lacks the Indian ethnicity including large standoff acquisition.
>
> Lastly, thanks for the suggestion of extending the dataset, that is the future we aim to not only extend the dataset in terms of the number of subjects but also to capture the images at varying distances.
>
> We will be more than happy to provide further information if needed to resolve the concerns. We want to thank all the reviewers for their constructive feedback.

---

> > ### Author Response · Authors · 2023-04-17
> > **References**
> >
> > [1] Multi-resolution face recognition with drones. In International Conference on Sensors, Signal and Image Processing 2020 (pp. 13-18).
> >
> > [2] Dronesurf: Benchmark dataset for drone-based face recognition. In IEEE International Conference on Automatic Face & Gesture Recognition (FG 2019) 2019 (pp. 1-7).
> >
> > [3] https://www.kaggle.com/datasets/jangedoo/utkface-new
> >
> > [4] https://github.com/abr-98/Face_data_based_deeplearning
> >
> > [5] https://github.com/dchen236/FairFace
> >
> > [6] https://github.com/dchen236/FairFace/blob/master/predict.py

---

### Meta-Review · Area_Chair_mBvs · 2023-04-08

**Recommendation:** Invite to revise
**Confidence:** 4

**Metareview:**

The paper presents a new dataset of individuals identifying as of Indian ethnicity for face attribute identification, and evaluates a soft biometrics predictor using the proposed dataset. The authors analyze the performance of an existing soft biometrics recognition network when tested in the wild on people identifying as of Indian ethnicity. The analysis reviews the known and important issue of bias towards the data distribution seen during training, which can heavily affect performance when tested on different data. The paper presents a detailed experimental analysis of deep face recognition algorithms and highlights the need for unbiased models for accurate face attribute analysis.

Review 1 notes that the paper lacks important details of the proposed dataset and suggests the evaluation of different recognition methods rather than different face detection models. The reviewer also questions the motivations provided in the abstract and the introduction and mentions the degraded performance of LightFace, which may be caused by the ethnicity of the participants or the low resolution of their faces. As future work, the authors could extend their dataset with recordings at different distances to assess how much the face resolution affects the overall performance.

Review 2 raises concerns about the ethical considerations of the work and suggests that there is no clarification provided on whether consent was obtained from the subjects. The reviewer also notes that the authors fail to make any comparisons with models trained on UTKFace or FairFace datasets that incorporate different races and identities. Moreover, there seem to be no experiments with current state-of-the-art models in gender and age estimation.

Review 3 notes that the authors curated a real-world dataset for Indian ethnicity individuals in unconstrained settings and presents a detailed experimental analysis of deep face recognition algorithms. The reviewer suggests that using biometric data such as facial images is restricted under GDPR, and there are other datasets available that include Indian ethnicity, such as the UTKFace dataset and the Adience dataset. The reviewer suggests increasing the sample size and using other available datasets to conduct the study.

Overall, the paper's main contribution is the proposal of a new dataset and the evaluation of a soft biometrics predictor using the proposed dataset. The paper highlights the importance of unbiased models for accurate face attribute analysis and presents a detailed experimental analysis of deep face recognition algorithms. However, the paper lacks important details of the proposed dataset, and there are ethical concerns regarding the collection of data. Moreover, the authors fail to make any comparisons with models trained on other datasets that incorporate different races and identities.


**Summary:**

The paper presents a new dataset of individuals identifying as of Indian ethnicity for face attribute identification, and evaluates a soft biometrics predictor using the proposed dataset.

**Reason For Not Giving A Higher Recommendation:**

The reviewers have raised many concerns that need to be addressed before this work can be considered CCR. Please see the weakness sections for details.


**Reason For Not Giving A Lower Recommendation:**

N/A

---

> ### Author Response · Authors · 2023-04-17
> **Revision**
>
> $\textit{Global:}$ We would like to thank the reviewers and meta-reviewers for appreciating the efforts taken to capture the dataset, along with highlighting the important issue of bias in soft-biometrics identification. Face recognition to identify individuals at large standoff is critical; therefore, the presence of our dataset can help in building robust face recognition algorithms in unconstrained settings. While in the literature, several studies have been conducted for face recognition, limited work has been done towards understanding the biasness in evaluating soft biometrics, including in the Indian context.
>
> $\textbf{R1:}$ We would like to thank the reviewer for highlighting this point. Here is the information R1 requested to provide: to avoid any biasedness in the evaluation, we have experimented with an equal gender ratio, i.e. 15 subjects are male, and 15 subjects are female. The age range of the subjects is between 19 and 34, with an average of 22.36 years. The videos are captured at a frame rate of 30.01 frames/rate, with a frame width and height of 1920 × 1080 pixels, and last for 5-7 seconds. The videos are captured in unconstrained indoor and outdoor environments where no special instructions are provided to the subjects and users are free to perform any natural motions.
>
> The motivation for utilizing the different face detection models is that the face is the critical component, and the effectiveness of the detection model significantly impacts the recognition performance[1-2]. Therefore, in this preliminary study, we evaluated several face detection models. However, based on the suggestion of the reviewer, we have now conducted an additional experiment with a face attribute classification network trained on face datasets, namely UTKFace [3], having balanced age, gender, and ethnicity images. The model trained on the UTKFace dataset [4] yields the F-1 score of 0.39, reflecting that even the models trained on a balanced dataset are ineffective under the unconstrained nature of the dataset. Interestingly, the performance is lower than the deepface model used in the paper, yielding an F-1 score of 0.43 for gender prediction.
>
> Further, we have also evaluated the face attribute prediction model [6] trained on another ethnicity-balanced dataset, namely Fairface [4]. Similar to the model trained on UTKFace, the model trained on Fairface is able to correctly predict the age of 200 images out of 450 within a tolerance of 10 years which is significantly lower than the deep face recognition system used in the paper, showcasing the biased behavior of existing models.
>
> Thanks for the suggestion of extending the dataset, that is the future we aim to not only extend the dataset in terms of the number of subjects but also to capture the images at varying distances. Further, the proposed dataset will be released to the public for research purposes, and one can receive the dataset after signing the license agreement.
>
> $\textbf{R2:}$ Thanks for raising this important concern. We want to highlight that the proposed dataset is collected by following strict ethical guidelines. Further, the subjects were informed that their biometric images are getting collected for research purposes and will not release any information related to identity. The dataset will be securely released to the biometrics community after signing the license agreement and will preserve the privacy of the subjects in the best possible manner.
>
> Based on the suggestion of the reviewer, we have now conducted an additional experiment with a face attribute classification network trained on face datasets, namely UTKFace [3], having balanced age, gender, and ethnicity images. The model trained on the UTKFace dataset [4] yields the F-1 score of 0.39, reflecting that even the models trained on a balanced dataset are ineffective under the unconstrained nature of the dataset. Interestingly, the performance is lower than the deepface model used in the paper, yielding an F-1 score of 0.43 for gender prediction. Further, when the UTKFace model is used for the age prediction with a tolerance limit of 10, the model correctly classifies 317 images, whereas the deepface successfully predicts 400 images.
>
> We have also evaluated the face attribute prediction model [6] trained on another ethnicity-balanced dataset, namely Fairface [5]. Similar to the model trained on UTKFace, the model trained on Fairface is able to correctly predict the age of 200 individuals only within a tolerance of 10 years which is significantly lower than the deep face recognition system used in the paper, showcasing the biased behavior of existing models.
>
> [1] Multi-resolution face recognition with drones. In ICSSIP 2020.
> [2] Dronesurf: Benchmark dataset for drone-based face recognition. In FG 2019.
> [3] https://www.kaggle.com/datasets/jangedoo/utkface-new
> [4] https://github.com/abr-98/Face_data_based_deeplearning
> [5] https://github.com/dchen236/FairFace

---

> > ### Author Response · Authors · 2023-04-17
> > **Revision (R3)**
> >
> > $\textbf{R3:}$ Thanks for pointing out the datasets. We will take a look at these datasets and will include them in our future studies. However, one important thing which we would like to highlight is that the proposed dataset does not only aim to handle Indian ethnicity but also aim to evaluate the effectiveness of existing deepface models for the images captured at unconstrained and large stand-off distance between subjects and camera. The above-mentioned datasets do not possess this feature (large unconstrained standoff distance).
> >
> > Additionally, our primary objective is to create a long-range distance dataset using a mobile phone, which is now accessible to almost everyone and is cost-effective. Currently, we are in the process of expanding our dataset size from 30 to 100-150 subjects and at varying standoff distances, which we have also pointed out in the future work of the paper.
> >
> > We will be more than happy to provide further information if needed to resolve the concerns. We want to thank all the reviewers for their constructive feedback.
> >
> > [1] Multi-resolution face recognition with drones. In International Conference on Sensors, Signal and Image Processing 2020 (pp. 13-18).
> > [2] Dronesurf: Benchmark dataset for drone-based face recognition. In IEEE International Conference on Automatic Face & Gesture Recognition (FG 2019) 2019 (pp. 1-7).
> > [3] https://www.kaggle.com/datasets/jangedoo/utkface-new
> > [4] https://github.com/abr-98/Face_data_based_deeplearning
> > [5] https://github.com/dchen236/FairFace

---

### Decision · Program_Chairs · 2023-04-10

Revision accepted; invite to archive

---

> ### Author Response · Authors · 2023-05-30
> **Archival Decision**
>
> We wish to opt-in for archival (publishing the paper).
>
> Thanks